# Time to Kill and Time to Heal: The Multifaceted Role of Lactoferrin and Lactoferricin in Host Defense

**DOI:** 10.3390/pharmaceutics15041056

**Published:** 2023-03-24

**Authors:** Anna Ohradanova-Repic, Romana Praženicová, Laura Gebetsberger, Tetiana Moskalets, Rostislav Skrabana, Ondrej Cehlar, Gabor Tajti, Hannes Stockinger, Vladimir Leksa

**Affiliations:** 1Institute for Hygiene and Applied Immunology, Center for Pathophysiology, Infectiology and Immunology, Medical University of Vienna, 1090 Vienna, Austria; 2Laboratory of Molecular Immunology, Institute of Molecular Biology, Slovak Academy of Sciences, 845 51 Bratislava, Slovakia; 3Laboratory of Structural Biology of Neurodegeneration, Institute of Neuroimmunology, Slovak Academy of Sciences, 845 10 Bratislava, Slovakia

**Keywords:** lactoferrin, lactoferricin, lactoferrampin, antimicrobial peptides, innate immunity, immunomodulation, inflammatory disease, COVID-19

## Abstract

Lactoferrin is an iron-binding glycoprotein present in most human exocrine fluids, particularly breast milk. Lactoferrin is also released from neutrophil granules, and its concentration increases rapidly at the site of inflammation. Immune cells of both the innate and the adaptive immune system express receptors for lactoferrin to modulate their functions in response to it. On the basis of these interactions, lactoferrin plays many roles in host defense, ranging from augmenting or calming inflammatory pathways to direct killing of pathogens. Complex biological activities of lactoferrin are determined by its ability to sequester iron and by its highly basic N-terminus, via which lactoferrin binds to a plethora of negatively charged surfaces of microorganisms and viruses, as well as to mammalian cells, both normal and cancerous. Proteolytic cleavage of lactoferrin in the digestive tract generates smaller peptides, such as N-terminally derived lactoferricin. Lactoferricin shares some of the properties of lactoferrin, but also exhibits unique characteristics and functions. In this review, we discuss the structure, functions, and potential therapeutic uses of lactoferrin, lactoferricin, and other lactoferrin-derived bioactive peptides in treating various infections and inflammatory conditions. Furthermore, we summarize clinical trials examining the effect of lactoferrin supplementation in disease treatment, with a special focus on its potential use in treating COVID-19.

## 1. Introduction

Breast milk acts not only as the first nutrition for infants but also as the first barrier against infections after birth. However, milk also offers health benefits beyond infancy. Casein-free whey protein fractions of milk contain cytokines, growth factors, antibodies, complement components, and various antimicrobial proteins, such as lactoferrin (LF) [1,2,3], the subject of this review.

The human glycoprotein LF (initially named lactotransferrin), a member of the transferrin family, is present not only in human milk, but also in exocrine secretions and body fluids (e.g., tears, saliva, and urine), in secondary granules of neutrophils, and on mucosal surfaces in the respiratory, urinary, reproductive, and intestinal tracts [4,5,6].

LF was discovered over 80 years ago [7]; since then, numerous studies have documented its activities and mechanisms of action in host defense. LF is particularly abundant in colostrum, the first form of breast milk produced after childbirth, which is why LF earned its nickname “newborns’ guardian”. By breastfeeding, infants receive among others the benefits of LF, which helps to protect them from infections until their own immune system fully develops [8].

LF found in the milk of cows, goats, sheep, buffaloes, and camels is commonly used as a nutritional supplement for older children and adults [9,10]. However, commercially purified LF produced on a large scale from animal milk is not only sold as a food supplement but also used in the pharmaceutical industry [4].

Upon ingestion, nutritional LF is proteolytically cleaved releasing natural bioactive peptides, such as lactoferricin (LFC), that not only retain some of the biological activities of LF but may also provide additional benefits [11].

Generally, LF is considered to be a multifunctional protein that plays a significant role in the immune system [12,13]. Its functions are diverse, including antibacterial, antifungal, antiviral, antiparasitic, antioxidant, antitumor, anti-inflammatory, and immunomodulatory activities [14]. These protective functions primarily depend either on the ability of LF to sequester iron ions or on the binding capacity of the positively charged N-terminal region, from which the natural peptide LFC is derived [14,15].

In neonatal medicine, LF and its derived natural or synthetic peptides are considered to be some of the most valuable and best-researched compounds [16], but they are also often used as supplements in multiple clinical trials for diarrheic, respiratory, inflammatory, dermal, or malignant diseases [17]. Recently, due to their antiviral potential, LF and its derivatives have also gained intensive attention as potential therapeutic tools against Coronavirus disease 2019 (COVID-19) [18].

## 2. Lactoferrin (LF)

### 2.1. Structure of LF

LF is a non-heme iron-binding glycoprotein that belongs to the transferrin family. All proteins from the transferrin family share a similar three-dimensional structure. Human LF is made of about 700 amino acids with a molecular weight of approximately 80 kDa (Figure 1). LFs from different species exhibit high homology [5,9,14,15]. A single polypeptide chain of LF is folded into two lobes (N and C) with 33–41% mutual homology [19], which probably originated from gene duplication [6,14]. LF has a less flexible structure than transferrin, with a relatively rigid linker formed by a three-turn helix, which connects the two lobes [20,21,22]. This linker may mediate a cooperative interaction between the LF lobes, contributing to its capacity to bind iron even at low pH [23]. The lobes are composed of α-helices and β-sheets assembled into two domains (N1 and N2 or C1 and C2 for the N- or C-lobe, respectively), with a metal-binding cleft in the middle. Each lobe, via its two tyrosines, one aspartate, and one histidine, can reversibly bind a single ferric (Fe^3+^) ion. The cleft can accommodate also other metal ions including ferrous iron (Fe^2+^), copper (Cu^2+^), zinc (Zn^2+^), manganese (Mn^2+^), aluminum (Al^3+^), or cerium (Ce^4+^) ions, implying that LF could also play a role in the homeostasis of these microelements [5,24]. However, the biological significance of binding of metals other than iron has not been investigated. Metal ions are bound in synergy with a carbonate ion CO_3_^2−^, which stabilizes them in the cleft through two oxygen atoms [5,20,21,24]. LF binds ferric iron with an extremely high affinity (Kd ~10^−20^ to 10^−22^ mol/L); affinities to other metal ions are markedly lower, and metal ion binding is accompanied by a conformational change within the protein structure [5,14,21,25]. Each lobe of LF can be either in the “closed” iron-bound state or in the “open” iron-free state [5,14]. Holo-LF with two bound iron cations displays a stable and conformationally rigid structure with iron ions difficult to remove. On the contrary, iron-free apo-LF is more flexible and prone to easier thermal denaturation and proteolysis. The release of iron depends on destabilization of the closed holo-form and is accompanied by a conformational change—domains in lobes swing away from each other to open the metal-binding clefts [5,25]. This is possible by lowering the pH, e.g., in the acidic compartments of cells (endosomes), leading to protonation of the carbonate ion, which results in weakening of the iron coordination to the point where it no longer holds the two domains together. Several available apo-LF X-ray structures display the closed configuration of the C-lobe, implying that open-close dynamics in solution might facilitate iron binding [5,14,22,25,26]. Furthermore, it has been observed that iron binding regulates human LF reactivity to glycan-binding proteins, suggesting that conformational changes accompanying iron chelation could be involved in the function of the molecule [27].

While LF and the prototype family member serum transferrin (TF) have the same global structure and use the same binding sites for binding iron, LF binds Fe^3+^ ions with 300-fold higher affinity than TF, and releases them at much lower pH values; LF was shown to retain iron to a pH as low as 3–4, compared to pH 5–6 for TF [5,23,26,28]. These properties set these two proteins functionally apart. Serum TF is primarily an iron transporter, whereas the main function of LF lies in efficient iron sequestration from the environment, via which LF contributes to innate immune defense and iron homeostasis (as explained in detail later) [6,14,22]. Further functional differences between LF and TF stem from the high number of basic (positively charged) residues contained in LF at its N-terminus (residues 1–7 and 13–30) and around the helix connecting the N- and C-lobes [5]. For this reason, LF is the most alkaline member of the transferrin family. The isoelectric point of LF is around 8.5–9.0 depending on the species, resulting in strong binding to many negatively charged extracellular, membrane-anchored, and intracellular molecules, such as DNA, lipopolysaccharide (LPS), or heparan sulfate proteoglycans (HSPGs) [5,6,14,18,29,30]. Lastly, LF is a glycoprotein with N-linked glycans, but the number and position of glycosylation sites vary between the mammalian species (Figure 1a) [5]. Human LF (hLF) has three, while bovine LF (bLF) has five potential N-glycosylation sites [5]. hLF contains core-fucosylated complex glycans (glycan antennas contain N-acetylglucosamine (GlcNAc), galactose, and sialic acid), whereas bLF contains both complex and high-mannose glycans (core GlcNAc dimer modified only with mannose residues) [31,32]. Glycosylation of both hLF and bLF found in milk exhibits dynamic changes over the course of lactation [32,33,34]. N-glycans do not influence LF binding of iron or LPS, but they protect it against tryptic proteolysis [5,35] and act as soluble decoy receptors for invasive pathogens that cause infections by binding to cells via their surface lectins, e.g., food-borne *Salmonella typhimurium* or *Listeria monocytogenes* [33,36]. LF glycosylation in multiple species was recently comprehensively reviewed by Zlatina and Galuska [31].

### 2.2. Tissue Distribution of LF

In situ hybridization and immunohistochemical analyses have revealed that LF is expressed during specific stages of murine embryogenesis. Its expression begins as early as the 2–4 cell fertilized embryo stage, continues until the blastocyst stage, and then becomes almost undetectable in the hatched blastocyst. This suggests that LF plays a role in preimplantation development. LF expression re-emerges later in gestation, specifically in neutrophils of the fetal liver and in epithelial cells of the respiratory and digestive systems [37]. There are no comprehensive reports on LF expression during human embryonic development yet. Only one report showed a positive correlation of LF concentrations in the follicular fluid with fertilization rate and embryo quality of in vitro fertilized patients, supporting the role of LF in the reproductive process in humans [38].

In adult humans, LF is secreted into mucosal fluids by glandular epithelial cells. It is constitutively present at the mucosal surface and regulated by various hormones and transcription factors in a tissue-specific manner [39,40]. Thus, LF is found in saliva, tears, semen, vaginal fluids, gastrointestinal fluids, nasal and bronchial secretions, and sweat [9,41,42]. However, the most enriched source of LF is breast milk with an approximate concentration of 2.6 mg/mL in human milk, 0.09 mg/mL in bovine milk, and the highest concentration in human colostrum with 5.3 ± 1.9 mg/mL [11,43,44,45,46]. 

Granulocyte colony-stimulating factor (G-CSF) induces expression of LF in secondary granules of neutrophils [47], where LF represents one of the major proteins with levels ranging from 3 to 15 µg/10^6^ neutrophils [48,49]. LF released from neutrophils occurs also in plasma, albeit at low concentration (0.2–1.5 µg/mL) [49]. Neutrophil-derived LF has also been detected in feces, where the concentration significantly rises during inflammation, e.g., as a response to pathogenic bacteria and in inflammatory bowel diseases (IBDs), such as ulcerative colitis and Crohn’s disease [50]. In this respect, fecal LF has been proposed as a specific and noninvasive marker to differentiate between IBD and noninflammatory irritable bowel syndrome [50,51]. Moreover, LF can be released from microglial cells, the resident macrophages in the brain [52]. Additionally, it was suggested that LF production might be triggered in some cell types by apoptosis [53].

### 2.3. Receptors of LF

LF performs various functions by binding to a wide range of receptors on target cells (Table 1), resulting in a vast array of cellular responses [6,54]. For instance, via binding the low-density lipoprotein receptor-related protein-1/alpha 2-macroglobulin receptor (LRP-1, CD91) on fibroblasts, LF promotes a phenomenon called collagen gel contraction, which is essentially the ability of fibroblasts to reorganize the surrounding 3D collagen matrix into a more dense and compact structure during wound healing [55]. By binding to LRP-1 on fibroblasts, LF also blocks LRP-1-dependent stimulation of cholesteryl ester synthesis elicited by β-very-low-density lipoprotein complexes [56]. Lastly, via LRP-1, LF has been shown to stimulate osteoblast proliferation [57]. Via C–X–C motif chemokine receptor 4 (CXCR4, CD184), LF induces the activation of AKT signaling in human epithelial cells [58]. In the intestine, LF and its derivatives have been shown to bind the glycan-binding lectin intelectin-1 (also known as omentin-1) for resorption of ingested LF and direct immunomodulation in the gastrointestinal tract [59,60]. LF was found to bind with high affinity to the soluble form of the LPS coreceptor CD14 [61]; via LPS, it was also shown to bind to Toll-like receptor 4 (TLR4) [62]. Moreover, interactions with TLR2 and endosomal TLR9 in macrophages and antagonism of their pattern recognition role have been reported, which resulted in LF-mediated suppression of the TLR2- and TLR9-mediated proinflammatory response against Epstein–Barr virus [63]. Furthermore, one of the C-type lectin receptors, termed dendritic cell-specific intercellular adhesion molecule-3-grabbing non-integrin (DC-SIGN, CD209) was identified as an interaction partner of LF in dendritic cells (DCs) [64]. LF has also been reported to bind to proliferating cells, including immune cells [65]. In this case, the cell surface-expressed nucleolin was identified as the LF receptor. This interaction, however, does not lead to signaling; instead, the nucleolin–LF complex is internalized through vesicles of the recycling-degradation pathway [66]. Lastly, as mentioned earlier, LF via its basic N-terminus binds to HSPGs expressed on the surface of multiple cell types [11,18,67,68].

LF has also been shown to interact with several soluble molecules, such as ceruloplasmin, osteopontin, or plasminogen, in the blood or other biological fluids [6,69,70,71].

**Table 1 pharmaceutics-15-01056-t001:** Cell surface receptors of LF and cell types in which they are expressed.

LF Receptor	Expressed by	Selected References
LRP-1 (CD91)	Multiple cell types (fibroblasts, osteoblasts, myeloid cells)	[55,56,57,72]
CXCR4 (CD184)	Leukocytes, epithelial cells, platelets	[58,73]
Intelectin-1 (omentin-1)	Intestinal epithelial cells	[59,60]
CD14	Monocytes, macrophages, neutrophils	[61,74]
TLR2 (CD282), TLR4 (CD284)	Myeloid cells, endothelial cells	[62,63,75]
DC-SIGN (CD209)	Myeloid cells (DCs, certain macrophage types)	[64,76,77]
Nucleolin	Proliferating cells	[66]
HSPGs	Broadly expressed on various cells and as extracellular matrix macromolecules	[11,67,68]

## 3. LF-Derived Bioactive Natural and Synthetic Peptides

Ingested LF can be cleaved by pepsin in the stomach, but also by trypsin and chymotrypsin in the small intestine [78,79,80]. A previous study showed that hLF is largely digested by the digestive system of adults [78]. bLF might be more stable in the human digestive tract, since it was reported that more than 60% of both the apo- and the holo-forms of bLF resisted proteolytic digestion by pepsin in the stomach of study participants [81]. However, the two clinical trials performed by the same group are not directly comparable; the first trial assessed a digestion of 5 g of recombinant hLF in the stomach and the small intestine of ileostomy patients over 24 h, while the second assessed the digestion of 4.5 g of bLF in stomachs of probands with a nasogastric tube for up to 30 min [78,81]. In any case, the situation is different in newborns, whose gastrointestinal tract is not yet fully developed and unable to extensively digest LF present in human or cow milk, since it was shown that both hLF and bLF recovered from feces of infants were only partially hydrolyzed and also retained the iron-binding capacity [8,82]. Regardless of the outcomes of these studies, the digestion results in the generation of antimicrobial peptides, namely, LFCs (Figure 1).

### 3.1. Lactoferricin (LFC) and Derived Peptides

The bioactive peptides can be released from LF by pepsin cleavage in the acidic environment of the stomach. In the literature, they are sometimes collectively termed LFCs, which is ambiguous and may be confusing. In this work, we reserve the term LFC for the longest peptide released after pepsin cleavage, whereas shorter peptides derived from LFC are specified by amino-acid numbering according to the LF sequence.

Both bovine LFC (bLFC) and human LFC (hLFC) were first identified in 1992 by proteolytic digestion of bLF and hLF, respectively, with pepsin in vitro [83]. Later, bLFC was confirmed as a breakdown product of bLF in the human stomach [79]. Both peptides are derived from the highly basic N-terminal region of LF, are resistant to further pepsin cleavage, and preserve many activities of LF or are even more potent than the parental LF [83,84,85,86].

Due to the shared antimicrobial properties of LF and LFC, the activity of intact LF is commonly explained as a specific action of its N-terminus from which the LFC peptide is derived [84]. However, nuclear magnetic resonance spectroscopy studies have shown that, in contrast to intact LF, wherein the LFC-encompassing sequence adopts a βαβα motif [5], the conformation of the free LFC peptides is radically different (Figure 1) [84,87]. LFCs also lack the iron-binding capacity [11].

bLFC is a 25-residue peptide (corresponding to residues 17–41 of bLF) and folds in aqueous solution into an amphipathic hairpin with two antiparallel β-strands and one intramolecular disulfide bond [88]. hLFC is twice as long as bLFC, consisting of the N-terminal 49 residues (i.e., residues 1–49) of mature hLF in a single continuous chain stabilized by two intramolecular disulfide bonds. In contrast to bLFC, hLFC is mostly disordered in aqueous solution but it shows a propensity of an α-helix in a membrane mimetic solvent (Figure 1) [83,84,87]. hLFC bears nine positive charges at neutral pH; four of them are located as the tetra arginine sequence at positions 2–5 [83,84]. Both bLFC and hLFC are able to adopt an amphipathic conformation in membrane mimetic solvents with separated hydrophobic (represented by tryptophan residues) and positively charged hydrophilic (represented by arginine residues) surface regions, which is a specific feature of antimicrobial peptides. The amphipathic properties of LFCs, and antimicrobial peptides in general, enable an interaction with lipidic molecules, as well as with negatively charged surfaces of Gram-negative and Gram-positive bacteria, fungi, parasites, viruses, and tumor cells; through binding to negatively charged patterns, they exhibit antibacterial, antifungal, antiparasitic, antiviral, and antitumor activity [11,46,84].

The extremely basic character is a specific feature of LFCs, and both the high net positive charge and the position of the cationic residues appear to be important for the afore-mentioned activities [84]. A stabilized secondary structure is also important, as the absence of the disulfide bond resulting in a disruption of the cyclic form of both bLFC and hLFC impairs antibacterial and antiviral activity [89,90,91,92]. Differences in length and amino-acid composition of bLFC and hLFC, both mirrored in their secondary structures, result, to some extent, in different effectiveness; bLFC is considered the more potent one out of the two peptides [84,89,93].

Most research has examined the antimicrobial properties of bLFC and hLFC, as well as synthetic peptides derived from them. However, a limited number of studies have also investigated LFCs from other species such as pig, mouse, goat, and camel [89,92,94,95,96,97]. All these LFCs seem to be active, but bLFC beats them with its superior bactericidal activity [89].

### 3.2. Lactoferrampin (LFA)

Additional bioactive peptides derived from LF are termed LFAs. LFA was first identified from bLF and synthesized chemically in 2004 [98]. Like LFC, LFA is derived from the cationic N-lobe, specifically bLF residues 268–284, and it possesses antibacterial and antifungal properties [98]. In 2005, the same research group described a more potent and slightly longer peptide (comprising residues 265–284; Figure 1a, right) [99]. As many antimicrobial peptides, LFA also adopts an amphipathic α-helical conformation in membrane mimetic solvents, at least at the N-terminus, but remains relatively unstructured at the strongly positively charged C-terminus that is essential for its antimicrobial activity [99,100,101]. However, the corresponding region of hLF does not possess antimicrobial activity. It can, however, be modified in vitro by increasing the net positive charge near the C-terminus, thus gaining this potency [101].

Notably, a synthetic fusion peptide of bLFC and bLFA, termed LFchimera, has higher antibacterial activity than the individual bovine bioactive peptides, even against multidrug-resistant bacteria and against strains that normally form biofilms [102,103,104,105,106]. LFchimera was also shown to have a superior parasiticidal effect against the enteric parasites *Giardia intestinalis* and *Entamoeba histolytica* [107,108], as well as a profound anticancer effect [93].

## 4. LF and LFC in Host Defense

Although scientific advances have led to large-scale production and widespread distribution of vaccines, but also antiviral and antimicrobial drugs, viruses and microorganisms still remain one of the major threats worldwide. The ever-increasing number of reports of pathogen resistance, as well as the emergence and re-emergence of epidemics, force the medical and scientific community to constantly search for new molecules with protective potential. Antimicrobial proteins and peptides have proven to be a promising alternative. LF and its derivatives, via various features, such as iron sequestration, pathogen membrane disruption, peculiar binding properties, modulation of immune cell activation and function, and protease blockade, are crucial molecular players in host defense processes, against viruses and bacteria in particular.

### 4.1. LF in Iron Homeostasis

Initially, LF was thought to play a role in intestinal iron uptake, transport, and delivery in both neonates and adults. However, these theories were disproven in 2003 when research on LF-knockout mice revealed that they were viable, fertile, and did not exhibit any abnormalities in their serum iron levels [60]. Today, LF is recognized as a guardian of iron homeostasis, as it efficiently scavenges toxic free iron from the environment [22,109]. Free iron ions released from necrotic cells in infected and/or inflamed tissue have a central role in oxidative stress due to induction of free radicals, i.e., reactive oxygen species (ROS) and reactive nitrogen species (RNS) [13,22]. In this scenario, the iron-scavenging ability of apo-LF released from the secondary granules of neutrophils functions as an antioxidant, protecting tissues from unwanted free iron-induced formation of ROS and damage to cell membranes, proteins, lipids, and nucleic acids [22,110,111]. Similarly, such scavenging of iron deposits in the brains of Alzheimer and Parkinson disease patients may be maintained by LF released by activated microglia [52,112].

In addition, via iron sequestering, LF reduces iron availability for iron-dependent pathogens, which contributes to host protection against infections via so-called nutritional immunity [113,114]. Because of this, LF is considered an important component of the innate immune defense system [5,6,14].

Interestingly, the iron-regulatory role of LF can be exploited therapeutically; iron-loaded holo-LF has been shown to efficiently deliver iron into cancer cells and sensitize them to radiotherapy and ferroptosis (an iron-dependent regulated cell death) via increased ROS production [115].

### 4.2. LF and LFC in Immunomodulation

Neutrophils play a crucial role in the immune response. During inflammation, neutrophils are attracted to the site of injury or infection in large numbers, more than any other type of white blood cells. They play a crucial role in killing invading bacteria by phagocytosis or by releasing antimicrobial peptides at the site of infection [116]. Additionally, they release a vast amount of LF from their secondary granules [12,117], and it was observed that released LF acts as an alarmin to attract antigen-presenting cells (monocytes, macrophages, and DCs) to the inflamed site [118]. In this respect, LF-mediated recruitment of DCs is especially important, as this creates the link to the adaptive immune responses [117].

LF is released from neutrophil secondary granules predominantly in its iron-free form (apo-LF), and, by scavenging free iron, it contributes to bacterial killing via iron deprivation [113]. Additionally, LF normalizes the redox status within the tissue as outlined above, which is beneficial, because, even though ROS and RNS production is intimately linked to immune cell activation, their excessive and prolonged generation impairs the inflammatory responses of several immune cell types, including macrophages, T cells, and B cells, and can even lead to cell death [119,120].

The most prominent immunomodulatory effects of LF are attributed to its highly basic N-terminal region formed by the positively charged arginine and lysine residues [5,18,29,30]. Via this region, LF interacts with a broad spectrum of binding partners, e.g., LPS from cell walls of Gram-negative bacteria [121], lipoteichoic acid (LTA) from cell walls of Gram-positive bacteria [122], DNA, including unmethylated CpG islands found in microbial DNA [123,124], glycosaminoglycans [30], heparin [125], and other negatively charged molecules encompassed in both pathogens and host cells. Many of these LF ligands belong to the pathogen-associated molecular patterns (PAMPs) that are sensed by pattern recognition receptors (PRRs), such as Toll-like receptors (TLRs) on the surface of immune and certain nonimmune cells. Early detection of PAMPs is crucial for the initiation of innate immune responses, the ultimate goal of which is pathogen elimination. For example, free LPS, a very potent endotoxin, binds to the serum protein termed LPS-binding protein (LBP). The LPS–LBP complex is then transferred to CD14, which accelerates LPS recognition by the TLR4/myeloid differentiation-2 (MD-2) complex [126]. LPS-triggered homodimerization of the TLR4/MD-2/LPS complexes leads to the assembly of specific adaptors (MyD88—myeloid differentiation factor 88 and TRIF—Toll/interleukin-1 receptor domain containing adaptor inducing interferon-β). This triggers a signaling cascade resulting in activation of various transcription factors such as NF-κB to induce expression of cytokines, e.g., tumor necrosis factor-alpha (TNF-α), interleukin-1β (IL-1β), IL-6, and type I interferons (IFNs) [126]. TLR4-mediated signaling also induces ROS and RNS in myeloid cells [119] and promotes DC maturation, thereby linking innate and adaptive immunity [126].

LF antagonizes this meticulously organized cascade on several levels to prevent excessive LPS signaling via TLR4 and, hence, to exert anti-inflammatory effects. Firstly, LF competes with LBP for LPS binding, and it prevents the interaction of LPS with CD14 [127], which results in reduced production of proinflammatory cytokines from LPS-stimulated monocytes and macrophages [128,129,130]. Secondly, LF interacts with both free soluble CD14 (sCD14) and the sCD14–LPS complex, thereby preventing LPS-triggered activation of endothelial cells via TLR4 and upregulation of endothelial adhesion molecules [61]. Thirdly, LF treatment leads to downregulation of TLR4 expression on the macrophage surface [130].

On the other hand, LF has been also shown to promote macrophage activation via TLR4-dependent and TLR4-independent pathways in certain in vitro studies [62,75,131]. This could be in part related to its “contamination” by LPS, as LF is a very potent scavenger and carrier of this endotoxin [12,13,131,132]. Further experiments with LPS-depleted LF revealed that hLF is able to mildly activate the TLR4 pathway due to recognition of LF carbohydrate chains, probably complexed with CD14. However, when cells were simultaneously treated with LPS and LPS-depleted hLF, cytokine production was again dampened compared to treatment with LPS alone. This suggests that hLF may have a role as a moderate activator of the immune system, while it effectively neutralizes the strong proinflammatory actions of LPS [75]. Accordingly, LF administration is protective even against lethal challenge in experimental endotoxemia (intravenous or intraperitoneal LPS injection) or bacteremia (infection with live *Escherichia coli*) in rodents [133,134,135,136,137,138]. LF treatment substantially reduced the serum levels of LPS-induced cytokines TNF-α, IL-6, IL-10 and nitric oxide (NO) [134], and effectively counteracted LPS- and *E. coli*-induced intestinal injury that otherwise manifested as diarrhea [133,137,138]. Protective effects of LF against damage to the intestinal mucosa were attributed to its capacity to attenuate increased epithelial inflammation and permeability due to disruption of epithelial tight junctions caused by LPS [139,140,141].

Regarding other immunomodulatory functions of LF, it was observed that LF triggered maturation of monocyte-derived DCs (moDCs) through TLR2 and TLR4, which supported antigen-specific T cell responses in coculture experiments with efficient differentiation of naïve T cells toward the Th1 type [118,138,142]. LF-matured DCs displayed an enhanced release of the inflammatory chemokines CXCL-8 and CXCL-10 and dampened production of IL-6, IL-10, and CCL-20. This effect was not due to LPS contamination of the recombinant hLF used in these experiments [142]. Moreover, it was recently observed that, in the presence of ssRNA, bLF and bLFC increased production of IFN-α by plasmacytoid dendritic cells (pDCs) that are crucial for the antiviral response. No such effect was observed upon pDC treatment by bLF or bLFC alone, suggesting that LF modulates the immune system by promoting pDC activation upon viral recognition [143]. However, when LF-treated moDCs were matured with LPS or other TLR ligands, a tolerogenic phenotype with decreased expression of activation markers, decreased release of proinflammatory cytokines, and decreased stimulatory capacity toward cocultured T cells was observed [144,145]. This again indicates that LF may represent a strategy to block excessive activation of the immune system upon inflammation triggered by TLR ligation.

LF also plays a role in balancing the potentially harmful excessive inflammatory response by acting as a negative feedback mechanism for neutrophils. It was observed that LF suppresses the release of neutrophil extracellular traps (NETs) both in vitro and in vivo [146]. NETs are a crucial defense mechanism against invasion of pathogens, but their uncontrolled release is associated with the development and progression of autoimmune diseases, inflammatory diseases, such as sepsis, and thrombosis. Exogenous hLF shrunk the chromatin fibers found in released NETs, without affecting the generation of ROS, but this failed after blockade of the positive charge in the N-terminus of LF with heparin, suggesting that charge–charge interactions between LF and NETs were required for this function [146].

These in vitro findings are supported by results of in vivo experiments in various disease animal models. They generally correlate with the predicted protective effects of LF. For example, in a mouse gastritis model induced by *Helicobacter pylori*, a Gram-negative pathogen linked to gastritis, ulceration, and stomach cancer, hLF treatment on top of the standard therapy dramatically contributed to *H. pylori* eradication, reduced gastric inflammation, and decreased the level of proinflammatory cytokines in stomach tissue [147]. Similarly, in a mouse peritonitis model using a lethal dose of the methicillin-resistant Gram-positive bacterium *Staphylococcus aureus* (MRSA), treatment with mouse LF resulted in modest increase in survival, albeit with reduced bacteremia and decreased serum levels of proinflammatory cytokines IL-17, IL-6, and IL-1β [97]. In a mouse model of tuberculosis, orally given bLF showed therapeutic effects against the aggressive *Mycobacterium tuberculosis* strain Erdman [148]. LF-treated mice showed reduced mycobacterial burden in the lung tissue, reduced lung inflammation with decreased foamy macrophages, and increased numbers of T cells that were producing IFN-γ and IL-17; these cells were thought to mediate protection against the infection. Furthermore, LF did not affect *M. tuberculosis* growth in vitro, but it enhanced the killing of mycobacteria by IFN-γ-stimulated macrophages in an NO-dependent manner [148]. In a follow-up study, orally given recombinant hLF or its fusion with the Fc domain of human IgG2 to extend its half-life in circulation were also protective against lung-induced pathology caused by the intravenous injection of the major cell wall component of *M. tuberculosis*, trehalose 6,6’-dimycolate (TDM) [149]. In this case, hLF and especially its Fc-fusion protein limited the lung inflammation and granuloma formation and contributed to resolution of the lung pathology over time. This was attributed to better control of TDM-induced proinflammatory cytokine production (TNF-α and IL-1β) in the lungs of mice treated by both forms of hLF [150]. LF also exhibited beneficial activities in experimental allergic rhinitis and allergic asthma, induced by sensitization with pollen or ovalbumin [151,152,153]. Protective effects of intranasally or orally applied hLF were attributed to lowering allergen-induced ROS levels in bronchial epithelial cells by apo-LF, with subsequent inhibition of eosinophil accumulation and reduced inflammation in the lungs [151], or to skewing of the immune response from the Th2 and Th17 to the protective Th1 type, as evident from increased airway IFN-γ and reduced Th2 (IL-4, IL-5, IL-13) and Th17 (IL-17) cytokines, as well as reduced ovalbumin-specific antibodies [152,153]. Parallel in vitro experiments revealed that LF influenced the phenotype of DCs and their antigen-presenting capacity, which resulted in a dampened ovalbumin-specific T cell response [153]. Lastly, the protective role of LF in allergy might be supported by a correlative observation that allergic rhinitis patients exhibit lower serum LF levels compared to nonallergic individuals [154].

Apart from microbial- or allergen-induced inflammation, several studies have shown that administration of LF may reduce experimental inflammation in tissues affected by other inflammatory disorders, such as neurodegenerative diseases [112], rheumatoid arthritis [155], IBD [156], or hyperoxia-induced lung and kidney inflammation [157]. In agreement with all these notions, LF knockout mice are more susceptible to various inflammatory disorders associated with impaired pathways of immune responses in various disease models [158,159,160,161,162].

Similarly to full-length LF, LFC was also found to convey immunomodulatory effects, i.e., by reducing the expression of proinflammatory cytokines and/or increasing the expression of anti-inflammatory cytokines [128,163,164,165,166]. In addition, it has also been shown that oral administration of the probiotic bacterium *Lactobacillus reuteri* expressing recombinant LFchimera to newborn piglets increases their resistance to enterotoxigenic *E. coli* (ETEC) strain K88 and diarrhea. LFchimera improves the piglet anti-inflammatory ability by inhibiting the NF-κB pathway, DC maturation, and T-cell activation, while also increasing their antioxidant capacity by activating the nuclear factor (erythroid-derived 2)-like 2 (NRF2)/heme oxygenase (HO-1) pathway, the master pathway that protects against oxidative damage caused by injury and inflammation [119,167]. Recently, it was confirmed that bLFC has anti-inflammatory properties by observing that it reduces the production of proinflammatory cytokines TNF-α and IL-6 in human and mouse macrophages upon stimulation with LPS. In particular, bLFC targeted the LPS-activated NF-κB and mitogen-activated protein kinase (MAPK) signaling pathways [168].

Lastly, it was shown that rectal administration of bLF did activate the bovine mucosal immune system and induce protection against enterohemorrhagic *Escherichia coli* (EHEC), likely through specific mucosal IgA [169]. Interestingly, the translocation of bLF into the nucleus of bovine rectal epithelial cells was observed in vitro upon inoculation with EHEC [170]. In this respect, it has been suggested that nucleolin together with proteoglycans mediates internalization of LF [66] (see also Section 2.3). Nevertheless, further analyses are needed to validate whether the nuclear translocation plays a role in the protective effects of bLF.

In summary, these studies showed that LF, LFC, and other LF-derived peptides display potent immunomodulatory functions to prevent infection- or inflammation-induced pathology (Figure 2). They predominantly target cells of the innate immune system (especially DCs, monocytes, and macrophages), which further influence cells of the adaptive immune system (T cells and B cells). However, very few studies pinpointed the action of LF to signaling through a particular LF receptor; so far, mainly actions via TLR4 have been studied [62,75,131,142]. Therefore, more research is needed to unravel the receptors and linked intracellular signaling pathways via which immunomodulatory actions of LF are executed.

### 4.3. LF and LFC as Inhibitors of Serine Proteases

Proteases exert essential functions in many physiological processes. Components of a plethora of proteolytic systems coordinate their activities to maintain homeostasis; yet, due to their enzymatic activities they can also contribute to a variety of human diseases. For example, the serine protease plasminogen is best characterized for its functions in fibrinolysis (resolution of blood clots) and cell migration. Plasminogen is also essential for immune cells to target pathogens or for endothelial cells to form vessels [171]. Nevertheless, human plasminogen might be also harnessed by malignant cancer cells and various pathogens. For instance, *Borrelia burgdorferi* hijacks plasminogen in a urokinase-type plasminogen activator (uPA)-dependent manner to increase invasiveness [172,173]. *Streptococcus pyogenes* secretes streptokinase capable of activating plasminogen independently of the host uPA [174]. Moreover, plasminogen and other serine proteases are implicated in proteolytic priming of various viruses [175]; the serine protease transmembrane protease serine type 2 (TMPRSS2) plays a decisive role in the proteolytic cleavage of the spike protein facilitating SARS-CoV-2 entry into host cells [176], while plasmin mediates the proteolytic processing of the H1N1 influenza virus hemagglutinin [177]. Hence, inhibitors of virus-priming serine proteases are a promising approach to manage virus infections.

In this regard, evidence has been gathered by us that LF directly binds plasminogen via its cationic N-terminus and reduces its proteolytic activity [71]. Furthermore, LF blocks the activation of plasminogen by *Borrelia* [71]. Moreover, LFC endows inhibitory potential toward plasminogen. Notably, LFC but not full-length LF is capable of blocking not only plasminogen conversion to plasmin but also the intrinsic plasmin proteolytic activity per se [178]. This difference between LF and LFC very likely stems from the above-described structural difference of free LFC from the N-terminal part within the whole molecule (see Figure 1). In addition, LFC blocks also TMPRSS2, which is homologous to plasminogen [178].

### 4.4. Direct Antiviral Activities of LF and LFC

Both hLF and bLF (and the respective LFCs, if tested) can block cell entry of many enveloped and naked viruses (Table 2) [11,179]. Through its N-terminal region, LF can bind to virus receptors (or coreceptors) on target mammalian cells, as well as block the initial stages of infection by several viruses, such as human immunodeficiency virus 1 (HIV-1) [58,64,73,76,180], dengue virus [181], herpes simplex virus 1 and 2 (HSV-1, HSV-2) [182,183], human and rat cytomegaloviruses (HCMV and RCMV), and several human coronaviruses (HCoVs, namely SARS-CoV-2, SARS-CoV-1, HCoV-229E, HCoV-NL63, and HCoV-OC43) [184,185,186]. In this respect, HSPGs as well-known and broadly expressed LF binding partners are of particular importance (Table 2) [30,54].

It has been observed that both bLF and bLFC are effective against HSV-1 and HSV-2 infection in vitro, with bLFC being more potent [182,183]. Furthermore, only bLFC administered therapeutically in a mouse model of HSV-2 genital infection was able to delay the onset of infection and significantly reduce the viral load, while bLF was not effective [183]. This suggests that the bioactive peptide might exert an antiviral activity via different mechanisms than the intact protein, due to differences in the 3D structure (see Figure 1). In particular, it was demonstrated that, although both LF and LFC can block viral entry via interaction with HSPGs, only LFC but not LF maintains its antiviral activity after the virus enters the cell [182]. This might be attributed to the specific properties of free LFC, such as the ability to enter the cell, as well as the higher capacity to induce an immune response in vivo [182,183].

To prevent human hepatitis C virus (HCV) infection, a direct binding of bLF or hLF to viral envelope proteins, rather than a blockade of a viral cell surface receptor is necessary [190]. Further research has shown that the interaction between the viral E2 protein and the C-terminal part of hLF, bLF, and horse LF was responsible for this effect [191].

Recently, LF has gained significant attention as studies have shown that LF and LF-derived peptides may inhibit the infection of SARS-CoV-2, the virus responsible for COVID-19, through the following mechanisms (Figure 3) [18,192]:
Through blocking the interaction between the viral S protein and HSPGs on the membrane of target cells by binding to HSPGs [184,185]. Interestingly, LF was observed to bind to HSPGs via its N-terminal region and inhibit cell infection of several HCoVs [185];Through direct binding to the S protein [189,193];Through blockade of the TMPRSS2-mediated virus priming [178]. This mechanism has been observed in particular for LFC, both synthetic and natural, but not for full-length LF, again pointing to differences between the released LFC and the corresponding region encompassed within the N-terminus of intact LF;Through blockade of the cathepsin L (CTSL)-mediated virus priming [194]. In this case, a bLF hydrolysate showed an inhibitory effect toward CTSL (a cysteine protease that primes SARS-CoV-2 in endosomes) that resulted in a decreased infection rate by a SARS-CoV-2 pseudovirus;Through enhancement of IFN responses [195]. It has been shown that bLF enhances the expression of IFN-β and downstream IFN-stimulated genes (e.g., MX1 and IFITM3), all of which are known to exert antiviral effects;Through inhibition of viral replication [196], which is due to direct inhibition of the viral RNA-dependent RNA polymerase (RdRp) activity by LF [197];Through maintenance of iron homeostasis [198];Possibly also through inhibition of the main viral protease M^pro^, also called 3CL^pro^ [199].

Furthermore, the antiviral activity of bLF and bLFC against several variants of SARS-CoV-2 has been studied in vitro [200]. In this case, bLF was more potent than bLFC, and the most profound effect of bLF was observed against the Alpha (B.1.1.7) variant. Different inhibition efficiencies of bLF onto individual SARS-CoV-2 variants are thought to be a result of their different affinities for HSPGs present on the surface of infected cells [200]. Taken together, these studies suggest that LF may be a potent, cost-effective, and widely available supplement in the management of COVID-19 due to its ability to target multiple stages of the virus life cycle.

### 4.5. Antibacterial, Antifungal, and Antiparasitic Activities of LF and LFC

LF has been shown to have a wide range of antibacterial effects (Figure 4), primarily due to its ability to sequester iron, as many pathogenic bacteria require iron for growth. These bacteriostatic properties of apo-LF are reversible, as bacterial growth can be restored by iron supplementation [11]. Nevertheless, the antimicrobial activities of LF are not only connected to iron deprivation [201]. LF has bactericidal effects due to binding to LPS, porins, and other outer membrane proteins of Gram-negative bacteria [121,202]. For example, LF binding to LPS causes LPS displacement and release, leading to the destabilization of the bacterial surface, depolarization of the outer bacterial membrane, and an increase in membrane permeability that ultimately results in enhanced bacterial susceptibility to osmotic shock, lysozyme, or antibiotics [203,204]. Although LF exerts its antimicrobial activities via two major skills, i.e., iron sequestration and selective binding, additional mechanisms via which LF impedes pathogens have been described.

It has been observed that lowering the iron concentration in the environment by LF inhibits bacterial biofilm formation by *Pseudomonas aeruginosa* by promoting a specific movement called twitching. This movement prevents the bacteria from attaching to mammalian cell surfaces, forming microcolonies and eventually a biofilm. A fivefold lower concentration of LF is needed to promote twitching than required for its bacteriostatic activity [205]. LF also inhibits biofilm formation of *Porphyromonas gingivalis* via its antiproteinase activity [206]. Furthermore, LF has been shown to impair the type III secretion system important for virulence of the gastrointestinal bacterial pathogen enteropathogenic *E. coli* (EPEC), a common cause of infant diarrhea in developing countries [207]. Recently, it was also reported that bLF decreased motility of the flagellated ETEC strain of *E. coli*, which led to reduced bacterial colonization of the small intestinal epithelium [208]. Due to these multiple activities, LF is considered as the primary defense protein against microbial infections [12].

Concerning bLFCs and derived synthetic peptides, they are more potent than bLF, and act swiftly and directly on the plasma membrane of bacteria, causing leakage as a result of membrane permeabilization [209,210,211]. The presence of hydrophobic tryptophan residues and charged arginine residues in bLFC were proven to be of particular importance for its antibacterial activity [212]. A very short peptide derived from bLFC (RRWQWR) belongs to the category of so-called cell-penetrating antimicrobial peptides. These peptides enter the bacterial cell without disrupting the membrane to execute their antimicrobial function by a still unknown mechanism [213]. Although hLFC and derived peptides are considered less potent than bLFC (and its derivatives) in terms of pore formation and/or depolarization of the bacterial membrane, they can still serve as a supplementary treatment to enhance the effectiveness of antibiotics against anaerobic biofilm microbes [214].

In addition, bLFC and its synthetic derivatives were recently extensively explored as antifungal [215] and antiparasitic agents [216]. Similarly, synthetic peptides derived from the N-terminus of hLF showed a spectrum of activities against bacteria and fungi in vitro, as well as in animal models with peroral administration of the peptides [96,217,218,219,220]. These peptides acted synergistically with other antibacterials and antifungals and contributed to a reduction in the minimum concentration for inhibition [217,218,221]. Of note, synthetic hLFC (1–11) was also effective against multidrug-resistant strains of *Staphylococcus aureus* in an animal model of chronic osteomyelitis, offering a possibility for an antibiotic-independent treatment of these difficult-to-treat infections [222]. Thus, LFCs and their derived synthetic peptides not only conserve some of the properties of LF but may also provide a synergetic potential and convey augmented antimicrobial effects, intrinsic to their altered conformations (Figure 1).

However, pathogens have developed host-specific strategies to neutralize the effects of LF and LFC. For example, *Streptococcus pneumoniae*, a Gram-positive bacterium and an obligate human pathogen, can bind hLF via its major virulence factor pneumococcal surface protein A (PspA). PspA has a stronger binding affinity for hLF than bLF [223]. PspA does not need to be attached to the bacterial surface to neutralize hLF, as soluble recombinant PspA can also bind hLF and protect the bacteria from the bactericidal activity of apo-hLF [223]. Using recombinant peptides, the hLF bactericidal capacity was mapped to two stretches of hLFC—peptides hLFC (1–11) and hLFC (21–31), but only the first peptide hLFC (1–11) was strongly neutralized by the soluble PspA, thus mapping the interaction of PspA to the very N-terminal part of hLF [223]. Another way in which pathogens can evade the effects of LF is through the synthesis of specific proteins that bind to holo-LF in order to extract iron and subvert the nutritional immunity. For instance, certain Gram-negative bacteria in the *Neisseriaceae* family utilize a two-component system, consisting of the outer membrane lipoprotein termed lactoferrin-binding protein B (LbpB) and the membrane transporter termed lactoferrin-binding protein A (LbpA). To extract iron from holo-hLF, LbpB interacts with hLF in a 1:1 ratio and delivers it to LbpA [54,224,225,226]. Furthermore, LbpB interaction with hLFC-derived synthetic peptide hLFC (1–11) protects the bacterium against its killing activity [227]. A recent structural study revealed that holo-hLF and hLFC (1–11) interact with LbpB in a noncompetitive manner; the C-lobe of hLF binds to the LbpB N-lobe, whereas hLFC (1–11) binds negatively charged patches in the LbpB C-lobe. Thus, LbpB allows the bacteria to execute two actions: to obtain iron for growth while neutralizing the antimicrobial hLFC peptide [226]. Recently, these bacterial proteins that neutralize LF have been targeted as a potential solution for vaccines against colonization and invasive infections. This strategy may also help to prevent the selection and emergence of strains that can steal iron from the host LF [228].

### 4.6. Antitumor Activities of LF and LFC

In addition to its antimicrobial and immunomodulatory activities, a plethora of antitumor properties have been ascribed to LF, including the inhibition of tumor cell growth and metastasis, as well as a protective effect against carcinogenesis [229,230].

LF may exert its antitumor activities directly through the inhibition of tumor cell proliferation, e.g., by induction of cell-cycle arrest at the G1/S transition through modulation of the expression of cell-cycle-regulatory proteins [231,232]. Recombinant hLF has also been shown to induce apoptosis-related morphological changes, disruption of the cytoskeletal structure, phosphatidylserine externalization, cell-cycle arrest, and selective cytotoxicity in triple-negative breast cancer cells [233]. In addition, LF has been found to inhibit signaling pathways that promote tumor growth, such as the MAPK and AKT pathways [229,234], and it can induce cancer cell apoptosis by increasing the expression of Fas death receptors and activating the Fas death-inducing signaling complex and the caspase cascade [235]. Furthermore, increased ROS production by bLF, leading to oxidative stress and subsequent apoptosis of human prostate and cervical cancer cells has been reported [236,237]. Several studies have also shown that LF inhibits tumor progression through suppression of tumor angiogenesis [238,239,240]. Various mechanisms have been described, one of them involving the inhibition of the TNF receptor-associated factor 6 (TRAF6)/NF-κB pathway by bLF in tumor endothelial cells but not in nontumor endothelial cells. This led to suppression of hypoxia-inducible factor 1 α (HIF-1α) activation, reduced production of vascular endothelial growth factor A (VEGF-A), and decreased growth of tumor blood vessels [239]. Another important antitumor and antimetastatic effect of LF has been recognized in its inhibitory capacity toward cancer cell migration [241,242,243]. For the observed effect, e.g., a reversal of the epithelial-to-mesenchymal transition (EMT) in LF-treated cancer cells by downregulating the EMT marker SNAIL and upregulating cadherins has been identified as an underlying mechanism [241,243].

Additionally, LF may affect tumor growth and spread indirectly via its immunomodulatory mechanisms, by stimulating lymphocyte and natural killer (NK) cell activity. For example, oral administration of hLF increased the numbers of tumor-infiltrating CD4^+^ and CD8^+^ T cells in a mouse model of head and neck squamous cell carcinoma [244]. In a breast cancer mouse model, orally administered hLF enhanced intestinal IFN-γ production, which was associated with expansion of NKT cells and CD8^+^ T cells, as well as increased systemic cytotoxicity of tumor-specific CD8^+^ T cells [245]. Similarly, in a mouse model of cervical carcinoma expressing hLF, impaired tumor growth was associated with an increase in CD4^+^ and CD8^+^ T lymphocytes in peripheral blood [246]. Furthermore, studies in melanoma-bearing mice showed that LF deficiency led to increased lung metastasis through recruitment of myeloid-derived suppressor cells to the lungs and suppression of TLR9 signaling, which were reversible by oral treatment with LF, indicating an antimetastatic effect of LF [230].

Both hLF and bLF, when administered orally to mice with squamous cell carcinoma or basal-like breast cancer tumors, were found to enhance the effectiveness of conventional chemotherapy drugs, such as cisplatin and tamoxifen, and more effectively control the tumor growth [247,248]. The improved antitumor effects of this combination therapy were attributed to the systemic immune-stimulatory effect of IL-18 and IFN-γ, which were upregulated by the LF-stimulated gut enterocytes [247,248].

Recently, by virtue of its versatility in cellular binding, LF has been recognized and used as a targeting ligand for disease-bearing cells. In particular, LF-conjugated nanoparticles and direct LF–drug conjugates have been used to carry chemotherapeutics into cancer cells to overcome cancer chemoresistance and halt tumor development in experimental animal models [249,250].

It is worth noting that LF has been shown to be downregulated in many tumor cell lines, animal models, and human cancer tissues, including breast, nasopharyngeal, and colorectal cancer [160,229,230,234]. LF deficiency has been associated with the development of inflammation-triggered carcinogenesis, as well as with enhanced tumor progression and metastasis [160,230,251], suggesting that LF may play a role as a tumor suppressor.

bLFC, hLFC and peptides derived from them have been shown to induce apoptosis in cancer cells by targeting outer leaflet-exposed plasma membrane phosphatidylserine [252], by inducing intracellular ROS and activating of Ca^2+^/Mg^2+^-dependent endonucleases [253], by activating the p53 signaling pathway and the caspase cascade [254], or by inhibiting autophagy at the late stage [255]. Additionally, bLFC has been reported to reverse cisplatin resistance in head and neck squamous cell carcinoma through downmodulation of PD-L1 expression [256].

## 5. LF in Clinical Trials

Over decades, LF and LFC have become well known for their multiple activities in host defense responses [11,12,13,18,46]. In addition to many in vitro and in vivo studies outlined above, LF has been examined in several clinical trials that focused on its potential application in the treatment of a variety of human diseases. This is possible because the European Food Safety Authority (EFSA, 2012) [257] and the US Food and Drug Administration (FDA, 2014; GRAS notice, GRN 465) recognized the use of bLF as a food supplement and as a part of infant formula as generally safe.

One of the most promising areas of LF usage in practice is neonatal medicine. Neonatal sepsis can have serious, even lethal consequences, and it poses a problem especially in developing countries. Preterm and/or very-low-birth-weight (<1500 g) infants are in particular danger [258]. Several studies have demonstrated that orally administered bLF plays a critical role in protecting newborns, especially preterm and low-birth-weight ones, from infections, bacterial and fungal sepsis, or necrotizing enterocolitis [259,260,261,262]. However, a similar clinical trial in very preterm infants performed in the United Kingdom did not find significant differences in the incidence of late-onset infection between bLF and placebo groups [263]. Another clinical trial in term infants who were formula-fed, conducted in Sweden, also found no significant differences in various immunological parameters or infection rates between infants whose formula was supplemented with bLF or not [264]. It was concluded that bLF supplementation was safe but with no additional benefit in this privileged population with low infection burden, and further studies in more vulnerable populations are warranted [264].

As a prevention or treatment against diarrhea, bLF or hLF has also been tested in several randomized controlled clinical trials with older probands. A clinical trial from Peru involving weaned toddlers, enrolled at 12–18 months and followed for 6 months, showed that oral bLF supplementation decreased longitudinal prevalence and severity of diarrhea over placebo, despite no decrease in diarrhea incidence [265]. A second clinical trial in Peru examined the effects of adding recombinant hLF and lysozyme to a rice-based oral rehydration solution for children aged 5–33 months with acute diarrhea. The results showed that the addition of hLF and lysozyme significantly reduced diarrhea duration and severity, as well as prevented new diarrhea episodes [266]. Additionally, a small clinical trial performed in long-term care elderly patients demonstrated that recombinant hLF was effective over placebo in prevention of antibiotic-associated diarrhea during an 8-week observation period [267].

In a clinical trial for *Helicobacter pylori* eradication, oral supplementation of bLF twice daily for 15 days on top of the standard treatment resulted in more efficient eradication of the pathogen in treated patients [268]. Some double-blinded randomized placebo-controlled clinical trials also focused on possible application of bLF in dental hygiene and the prevention of periodontitis. For example, a long-term daily intake of tablets containing high-dose bLF (60 mg/day) in combination with lactoperoxidase and glucose oxidase for 12 weeks was associated with an improvement in oral health-related quality of life and prevention of periodontitis in healthy adults [269].

LF may serve as a prophylactic agent against respiratory tract infections. Several randomized clinical trials, recently summarized in the meta-analysis by Ali et al. [270], found that a dietary intake of bLF, as fortified formula, oral dietary supplement, or oral gel, was associated with decreased incidence of upper- or lower-respiratory tract infections in infants, children, or adult subjects.

In oncology, a blinded randomized controlled clinical trial evaluated the effects of oral bLF on the growth of colorectal polyps, likely to be adenomas, over a period of 1 year [271,272]. The results showed that the highest dose of bLF halted the growth of polyps. Furthermore, participants with regressing polyps showed increased NK cell activity, elevated serum hLF levels, indicative of increased neutrophil activity, and higher numbers of polyp-infiltrating CD4^+^ T cells. These findings suggest that bLF suppresses colorectal polyps by enhancing immune responses [272]. Recently, a double-blind placebo-controlled trial demonstrated the benefit of oral bLF supplementation during induction chemotherapy in pediatric patients with hematologic malignancies, associated with promotion of gut microbiome homeostasis [273]. LF supplementation also seems to mitigate taste and smell abnormalities among cancer patients receiving chemotherapy, which are caused by chemotherapy-induced lipid peroxidation effects in the oral cavity [274].

bLF also seems to have a potential as a supplementary treatment of iron deficiency and iron deficiency anemia in both pregnant and nonpregnant women of childbearing age [275], as well as of iron deficiency anemia in children with IBD [276]. Treatment with bLF (with iron saturation, if stated, of about 30%) was well tolerated and more effective than treatment with ferrous sulfate in restoring iron homeostasis in both anemic women and children. This resulted in improvement of all hematological parameters measured (hematocrit, hemoglobin, total serum iron, and serum ferritin) and normalization of serum levels of IL-6 and of hepcidin, the central regulator of systemic iron homeostasis [275,276].

Last, but not least, LF has gained profound attention as a potential supplemental treatment for COVID-19. In this respect, two small studies were performed in Italy. In the first nonrandomized retrospective study, asymptomatic to moderately symptomatic COVID-19 patients who took bLF as a nutraceutical, or not, were followed up. Faster viral clearance and more pronounced reduction in symptoms with increasing patient age were observed in the group with oral bLF supplementation [277]. In the second trial, asymptomatic to mild-to-moderate COVID-19 patients were assigned to three groups: a mix of hospitalized and home-treated patients received oral and intranasal liposomal bLF, the second hospitalized group received standard-of-care treatment, and the third group of home-treated patients did not take any medication. It was reported that liposomal bLF treatment was without adverse effects and led to faster recovery with decreased serum ferritin, IL-6, and D-dimers, compared to standard-of-care-treated patients [278]. These studies suggest that LF might be beneficial as a supplementary therapy against COVID-19. In contrast, a recently published double-blinded randomized control trial found no link between bLF supplementation and prevention of COVID-19 in healthcare workers. In a few participants who got infected and took bLF versus placebo, no significant differences in severity and duration of symptomatic infection were observed [279]. Similarly, a small randomized prospective interventional pilot study from Egypt showed no statistically significant differences regarding recovery of symptoms or improvement of several laboratory parameters measured in mild-to-moderate COVID-19 patients who took supplemental LF or not [280]. In conclusion, larger randomized clinical trials focused on differential timing, dosage, and duration of the treatment with LF are needed to validate the potential beneficial roles for supplemental treatment of COVID-19.

## 6. Conclusions

LF is a multifaceted protein that plays a critical role in host defense. Its unique structural properties, such as the ability to sequester iron and the selective binding via its highly basic N-terminus, allow it to function as both a component of innate immune responses and a modulator of adaptive responses. By virtue of these capabilities, LF performs a wide range of immune functions, ranging from direct killing of pathogens to modulation of inflammation. Similarly, LFCs of various origin can bind to various targets, such as negatively charged lipids in the membranes of bacteria and lower eukaryotes or the outer leaflet of cancer cells. Thus, both LF and LFC are highly valuable natural substances with a range of potential health benefits that can balance between pathogen elimination and promoting healing, in line with the statement in Ecclesiastes 3:3, when it is “a time to kill and a time to heal”.

## Figures and Tables

**Figure 1 pharmaceutics-15-01056-f001:**
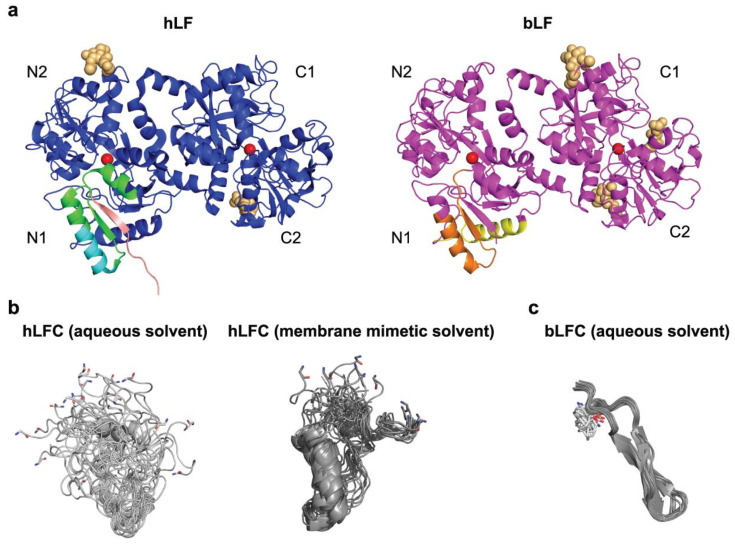
Domain architecture of human (hLF) and bovine (bLF) lactoferrin and the localization of the bioactive peptides within. (**a**) X-ray structures of holo-hLF (PDB ID 2BJJ, blue cartoon) and holo-bLF (PDB ID 1BLF, magenta cartoon). The positions of individual domains in the N- and C-lobes are indicated. hLFC (residues 1–49) is highlighted in different colors (green, salmon, and cyan) in the N1 domain of hLF. In particular, stretches of hLFC carrying a pronounced antimicrobial activity, often studied in the form of synthetic peptides (hLFC (residues 1–11) and hLFC (residues 21–31)), are shown in salmon and cyan, respectively. bLFC (residues 17–41) and bovine lactoferrampin (bLFA) (residues 265–284) are shown in orange and yellow, respectively, in the N1 domain of bLF. Iron ions bound to bLF and hLF are shown as red spheres. Note the differences in the glycosylation pattern of both proteins (asparagine and the first N-acetylglucosamine sugar residue are shown as beige spheres). (**b**) Twenty NMR solution structures of hLFC in aqueous solution (PDB ID 1Z6W) and in a membrane mimetic solvent (PDB ID 1Z6V). hLFC adopts a mostly disordered coiled conformation in aqueous solution. The flexibility of this conformation facilitates an adaptive binding to negatively charged surfaces of pathogen proteins. In a membrane mimetic solvent, the central part of hLFC acquires an amphipathic helical conformation, similar to the one it adopts in intact LF, thereby acting to disrupt pathogen membranes. (**c**) Twenty NMR solution structures of bLFC in aqueous solution (PDB ID 1LFC, gray). In contrast to the αβ conformation that retains the same residues in intact bLF (as shown in panel (**a**)), the bLFC peptide has a strong tendency to form a distorted, amphipathic antiparallel β-sheet. The N-terminal glycine residues in (**b**) and the N-terminal phenylalanine residues in (**c**) are highlighted as sticks; with carbon, nitrogen, and oxygen atoms labeled in grey, blue, and red, respectively. All structures were generated using PyMOL software.

**Figure 2 pharmaceutics-15-01056-f002:**
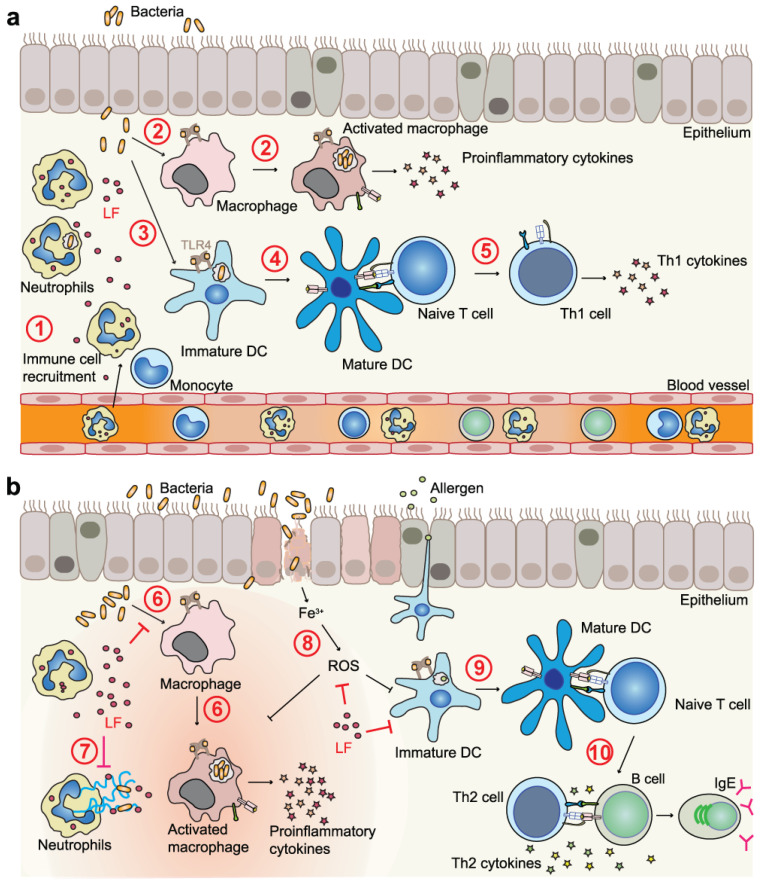
Modulation of cells of the innate and adaptive immune system by LF contributes to host defense. (**a**) In early phases of the immune response to invading pathogens, LF released from neutrophil granules has an immunostimulatory role because of its function as an alarmin to mediate immune cell recruitment from the bloodstream to the affected tissue (1) and by activation of myeloid cells—macrophages (2) and DCs (3) in particular. These LF actions might be in part related to its ability to serve as a PAMP carrier. Lastly, by promoting DC maturation (4), LF has a positive influence on functions of the adaptive immune system by activating, for instance, T cells (5). (**b**) In contrast, in chronic infection or inflammation (e.g., in allergic inflammation), LF exerts immunomodulatory functions and, therefore, protective effects via several mechanisms. LF efficiently counteracts the excessive proinflammatory responses of macrophages triggered by PAMPs (6) and shrinks neutrophil extracellular traps (7). LF also efficiently scavenges free ferric iron, thereby blocking deleterious ROS production (8). In allergy, LF is known to modulate DC functions (9), thereby skewing the immune response from the harmful Th2 to the protective Th1 type (10) that is accompanied by reduced production of allergen-specific IgE antibodies by B cells.

**Figure 3 pharmaceutics-15-01056-f003:**
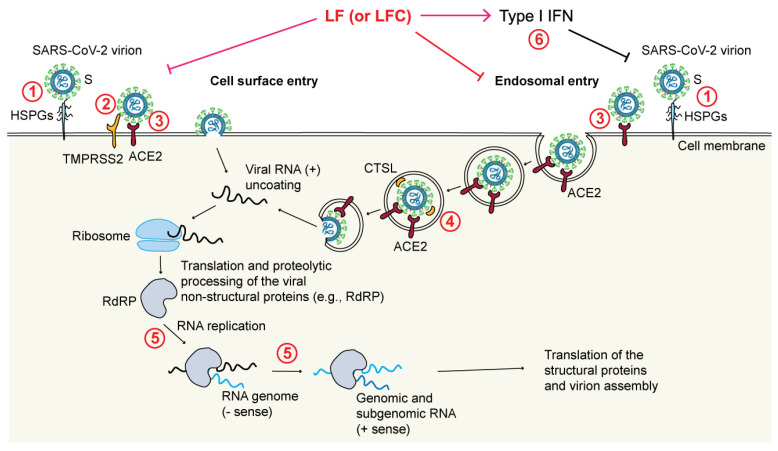
LF and LFC can interfere with SARS-CoV-2 infection through various mechanisms. SARS-CoV-2 enters cells via two different routes: (i) via the cell surface, mediated by binding to HSPGs and ACE2 with S protein processing by the cell surface serine protease TMPRSS2, which is required for membrane fusion and subsequent infection, or (ii) via endosomes, in which HSPGs, ACE2, and the endosomal cysteine protease CTSL play a role. LF has been shown to interfere with both pathways, due to binding to HSPGs (1) or directly to the S protein (3), and due to inhibition of CTSL that cleaves the S protein to release the virus from the endosome (4). Furthermore, LF has been found to inhibit viral replication by directly targeting the viral RNA-dependent RNA polymerase (RdRP, 5). Lastly, LF can boost the antiviral response through enhancing type I IFNs (6). So far, LFC has been shown to inhibit viral entry by inhibition of the protease TMPRSS2 (2).

**Figure 4 pharmaceutics-15-01056-f004:**
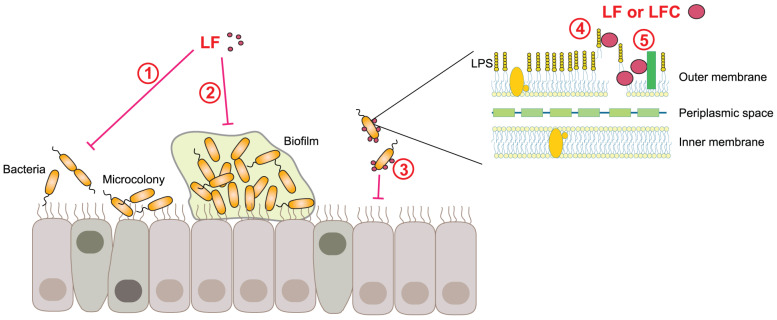
Antimicrobial actions of LF and LFC. LF has antibacterial activity toward a spectrum of different bacterial pathogens, through iron sequestration, which is bacteriostatic (1), or leads to bacterial twitching that prevents biofilm formation (2). Moreover, via its N-glycans, LF acts as a soluble decoy receptor for invasive pathogens to disrupt their adherence to host cells and cell invasion strategies (3). LF and LFC can cause membrane permeabilization (4) and bind and neutralize bacterial virulence mechanisms, e.g., the type III secretion system of EPEC (5).

**Table 2 pharmaceutics-15-01056-t002:** Role of LF and LFC in inhibition of virus infection by targeting the virus or viral (co)receptors.

Virus	Mechanism of Action	Selected References
HIV-1	Blockade of viral coreceptor CXCR4 of CXCR4-tropic HIV-1 strain by hLF	[58,73]
	Inhibition of HIV-1 transfer from DCs to CD4 T cells by blockade of DC-SIGN (CD209) by bLF or hLF (and partially also by hLFC)	[64,76,180]
HCMV, RCMV	Blockade of viral cell entry by hLF, bLF, hLFC, or bLFC, likely via interaction with HSPGs on target cells	[92,187]
Dengue virus	Blockade of viral coreceptors HSPGs, LRP-1 (CD91) and DC-SIGN (CD209) by bLF	[181]
HSV-1, HSV-2	Blockade of HSPGs by hLF, bLF, hLFC, or bLFC; LFCs are more potent than LFs because of other mechanisms employed	[90,182,183]
Zika and Chikungunya viruses	Blockade of viral cell entry by bLF, probably via interaction with HSPGs on target cells	[188]
SARS-CoV-2 and other HCoVs	Inhibition of SARS-CoV-2, SARS-CoV-1, and other HCoV attachment because of the HSPG blockade by hLF or bLF	[184,185,186]
	Blockade of SARS-CoV-2 S protein by hLF or bLF	[189]
HCV	Direct blockade of HCV virions by bLF and hLF	[190]

## Data Availability

The data presented in this study are available in this article.

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
