# Peer review of "Time to Kill and Time to Heal: The Multifaceted Role of Lactoferrin and Lactoferricin in Host Defense"

_pharmaceutics, 2023, doi:10.3390/pharmaceutics15041056_

Round 1

Reviewer 1 Report

The manuscript titled "Time to Kill and Time to Heal: The Multifaceted Role of Lactoferrin and Lactoferricin in Host Defense" is a review where the functional and structural characteristics of Lactoferrin were addressed.

The topic addressed in this review is very broad, with no depth in specific subtopics. If appropriate, the authors could deepen with more information on the topic "LF in Clinical Trials", as reported in the review the application of LF in the treatment of patients with COVID-19.

This LF review bears little resemblance to this publication "The Biology of Lactoferrin, an Iron-Binding Protein That Can Help Defend Against Viruses and Bacteria, Front. Immunol., 28 May 2020
sec. Viral Immunology Volume 11 - 2020 | https://doi.org/10.3389/fimmu.2020.01221", however, as it is a review, some topics end up being similar, but the approach and the complement of information justifies the publication.

I suggest authors include an explanatory figure on the mode of action of LF on microorganisms.

The conclusion, and bibliographical references are in agreement.

Author Response

We thank Reviewer #1 for the valuable comments. Below, we respond in a point-by-point fashion:

The manuscript titled "Time to Kill and Time to Heal: The Multifaceted Role of Lactoferrin and Lactoferricin in Host Defense" is a review where the functional and structural characteristics of Lactoferrin were addressed.
The topic addressed in this review is very broad, with no depth in specific subtopics. If appropriate, the authors could deepen with more information on the topic "LF in Clinical Trials", as reported in the review the application of LF in the treatment of patients with COVID-19.

Response: As suggested by the reviewer, we broadened the section describing the use of lactoferrin in clinical trials for treatment of other diseases. For details, please see the lines 856-866 of the revised manuscript.

This LF review bears little resemblance to this publication "The Biology of Lactoferrin, an Iron-Binding Protein That Can Help Defend Against Viruses and Bacteria, Front. Immunol., 28 May 2020 sec. Viral Immunology Volume 11 - 2020 | https://doi.org/10.3389/fimmu.2020.01221", however, as it is a review, some topics end up being similar, but the approach and the complement of information justifies the publication.

Response: We can assure Reviewer #1 that the resemblance to the Kell et al. review is purely accidental. The general structure of the two manuscripts might be similar, since we, as well as Kell et al, attempted to provide the complete overview on lactoferrin, including the structure, receptors, tissue and cellular localization and the mechanisms of action as an antimicrobial, and antiviral molecule. Both reviews also devote quite a lot of space to SARS-CoV-2 and COVID-19, but this again is justified by the fact that both Kell et al., and we are active in the SARS-CoV-2-related research.

However, there are also notable differences. Our review goes into more depth, especially when it comes to immunomodulatory functions of lactoferrin and lactoferricin. Additionally, only we focus on various bioactive peptides derived from lactoferrin (lactoferricin, lactoferramin, LFchimera), as well as on anticancer properties of lactoferrin and lactoferricin, which are completely missing from the Kell et al. review. For this reason, we are relieved that the reviewer justifies our approach and has no objection to the publication of our manuscript. Nevertheless, we critically compared the two reviews and are citing now the work of Kell et al. as a reference 54 at the most appropriate sites.

I suggest authors include an explanatory figure on the mode of action of LF on microorganisms.

Response: We thank the reviewer for the valuable suggestion. We included the figure, which now appears as Figure 4 of the updated manuscript.

The conclusion, and bibliographical references are in agreement.

Response:
We thank the reviewer for overall positive evaluation of our manuscript.

Reviewer 2 Report

It's quite a long review on lactoferrin, and it doesn't go into much of the glycan conformation of the protein.

It is important given the relevance that glycans may have in the development of several of their functions, since very little is mentioned in this regard in the manuscript, so I recommend that this topic be studied further and added in this manuscript.

The anticancer activity of lactoferrin and its peptides has also been extensively studied, in different types of cancer, including breast cancer and leukemia, and much information is lacking on the cancer review. Consult the following articles to enrich this topic:

Iglesias-Figueroa, B. F., Siqueiros-Cendón, T. S., Gutierrez, D. A., Aguilera, R. J., Espinoza-Sánchez, E. A., Arévalo-Gallegos, S., ... & Rascón-Cruz, Q. (2019). Recombinant human lactoferrin induces apoptosis, disruption of F-actin structure and cell cycle arrest with selective cytotoxicity on human triple negative breast cancer cells. Apoptosis, 24(7-8), 562-577.

Lu, Y., Zhang, T. F., Shi, Y., Zhou, H. W., Chen, Q., Wei, . Y., ... & Fu, C. Y. (2016). PFR peptide, one of the antimicrobial peptides identified from the derivatives of lactoferrin, induces necrosis in leukemia cells. Scientific reports, 6(1), 20823.

Author Response

We thank Reviewer #2 for the valuable comments. Below, we respond in a point-by-point fashion:

It's quite a long review on lactoferrin, and it doesn't go into much of the glycan conformation of the protein.

It is important given the relevance that glycans may have in the development of several of their functions, since very little is mentioned in this regard in the manuscript, so I recommend that this topic be studied further and added in this manuscript.

Response: We agree with Reviewer #2. Our manuscript is rather lengthy, but we wished to provide a comprehensive overview on various aspects of lactoferrin (and lactoferricin) biology, with emphasis on their immunomodulatory functions and their effects in COVID-19. For this reason, lactoferrin decoration with N-glycans was only briefly mentioned in the first version of our manuscript. But as suggested by the reviewer, we enriched the part describing the lactoferrin glycosylation; for details, please see the lines 145-155 of the updated manuscript.

The anticancer activity of lactoferrin and its peptides has also been extensively studied, in different types of cancer, including breast cancer and leukemia, and much information is lacking on the cancer review. Consult the following articles to enrich this topic:

Iglesias-Figueroa, B. F., Siqueiros-Cendón, T. S., Gutierrez, D. A., Aguilera, R. J., Espinoza-Sánchez, E. A., Arévalo-Gallegos, S., ... & Rascón-Cruz, Q. (2019). Recombinant human lactoferrin induces apoptosis, disruption of F-actin structure and cell cycle arrest with selective cytotoxicity on human triple negative breast cancer cells. Apoptosis, 24(7-8), 562-577.

Lu, Y., Zhang, T. F., Shi, Y., Zhou, H. W., Chen, Q., Wei, . Y., ... & Fu, C. Y. (2016). PFR peptide, one of the antimicrobial peptides identified from the derivatives of lactoferrin, induces necrosis in leukemia cells. Scientific reports, 6(1), 20823.

Response: We thank the reviewer for the suggestion. We updated the section describing the anticancer activity of lactoferrin and its derived peptides. For details, please see lines 759-762, 766-768, 774-778 and 815-817 of the updated manuscript.

Reviewer 3 Report

The paper by Ohradanova-Repic A et al., entitled “Time to kill and time to heal: the multifaceted role of Lactoferrin and Lactoferricin in host defense” is a review of the existing literature concerning the different functional aspects of lactoferrin in the immune defense. The topic is of interest and is well developed by the authors as it mentions the different roles of lactoferrin described in the Literature. In fact, the authors describe in a sufficiently extensive manner and with the appropriate Literature, the phylogenetic origin, the chemical structure of the molecule and its derivative peptides, the mechanisms of action as an antimicrobial peptide, its cellular and tissue localization, the activity immunomodulator, escape mechanisms of microorganisms, and, notably, does not fail to mention clinical trials. The text is easy to read and the figures are sufficiently informative.

Author Response

We thank Reviewer #3 very much for reviewing our manuscript and for such a positive evaluation.